# Simultaneous Machine Translation with Tailored Reference

**Shoutao Guo** [1,2], **Shaolei Zhang** [1,2], **Yang Feng** [1,2*]

[1]Key Laboratory of Intelligent Information Processing,
Institute of Computing Technology, Chinese Academy of Sciences (ICT/CAS)
[2] University of Chinese Academy of Sciences, Beijing, China
guoshoutao22z@ict.ac.cn, zhangshaolei20z@ict.ac.cn, fengyang@ict.ac.cn

## Abstract

Simultaneous machine translation (SiMT) generates translation while reading the whole source sentence. However, existing SiMT models are typically trained using the same reference disregarding the varying amounts of available source information at different latency. Training the model with ground-truth at low latency may introduce forced anticipations, whereas utilizing reference consistent with the source word order at high latency results in performance degradation. Consequently, it is crucial to train the SiMT model with appropriate reference that avoids forced anticipations during training while maintaining high quality. In this paper, we propose a novel method that provides tailored reference for the SiMT models trained at different latency by rephrasing the ground-truth. Specifically, we introduce the tailor, induced by reinforcement learning, to modify ground-truth to the tailored reference. The SiMT model is trained with the tailored reference and jointly optimized with the tailor to enhance performance. Importantly, our method is applicable to a wide range of current SiMT approaches. Experiments on three translation tasks demonstrate that our method achieves state-of-the-art performance in both fixed and adaptive policies[1].

## 1 Introduction

Simultaneous machine translation (SiMT) (Gu et al., 2017; Ma et al., 2019, 2020) generates the target sentence while reading in the source sentence. Compared to Full-sentence translation, it faces a greater challenge because it has to make trade-offs between latency and translation quality (Zhang and Feng, 2022a). In applications, it needs to meet the requirements of different latency tolerances, such as online conferences and real-time subtitles. Therefore, the SiMT models trained at

---

[*]Corresponding author: Yang Feng.
[1]Code is available at https://github.com/ictnlp/Tailored-Ref

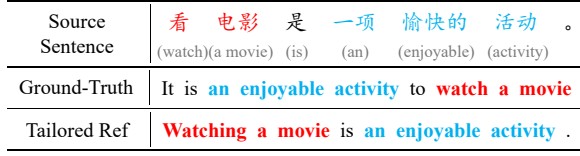

| Source Sentence | 看 (watch) | 电影 (a movie) | 是 (is) | 一项 (an) | 愉快的 (enjoyable) | 活动 (activity) | 。 |
|---|---|---|---|---|---|---|---|
| Ground-Truth | It is **an enjoyable activity** to **watch a movie** | | | | | | |
| Tailored Ref | **Watching a movie** is **an enjoyable activity** . | | | | | | |

Figure 1: An example of Chinese-English parallel sentence. The SiMT model will be forced to predict 'an enjoyable activity' before reading corresponding source tokens. In contrast, the tailored reference avoids forced anticipations while maintaining the original semantics.

different latency should exhibit excellent translation performance.

Using an inappropriate reference to train the SiMT model can significantly impact its performance. The optimal reference for the SiMT model trained at different latency varies. Under high latency, it is reasonable to train the SiMT model with ground-truth since the model can leverage sufficient source information (Zhang and Feng, 2022c). However, under low latency, the model is constrained by limited source information and thus requires reference consistent with the source word order (Chen et al., 2021). Therefore, the SiMT model should be trained with corresponding appropriate reference under different latency.

However, the existing SiMT methods, which employ fixed or adaptive policy, commonly utilize only ground-truth for training across different latency settings. For fixed policy (Ma et al., 2019; Elbayad et al., 2020; Zhang and Feng, 2021), the model generates translations based on the predefined rules. The SiMT models are often forced to anticipate target tokens with insufficient information or wait for unnecessary source tokens. For adaptive policy (Ma et al., 2020; Miao et al., 2021; Zhang and Feng, 2022b), the model can adjust its translation policy based on translation status. Nevertheless, the policy learning of SiMT model will gradually adapt to the given reference (Zhang et al., 2020). Consequently, employing only ground-truth

for the SiMT models trained at varying latency levels can negatively impact overall performance, as it forces them to learn the identical policy. Furthermore, Chen et al. (2021) adopts an offline approach to generate reference using the Full-sentence model for training the SiMT model at different latency, but this approach also imposes an upper bound on the performance of the SiMT model. Therefore, it is necessary to provide high-quality and appropriate reference for the models with different latency.

Under these grounds, we aim to dynamically provide an appropriate reference for training the SiMT models at different latency. In SiMT, the source information available to the translation model varies with latency (Ma et al., 2019). Therefore, the appropriate reference should allow the model to utilize the available information for predicting target tokens accurately. Otherwise, it will result in forced anticipations, where the model predicts target tokens in reference using insufficient source information (Guo et al., 2023). To explore the extent of forced anticipations when training the SiMT model with ground-truth at different latency, we introduce anticipation rate (AR) (Chen et al., 2021). As shown in Table 1, the anticipation rate decreases as the SiMT is trained with higher latency. Consequently, the reference requirements of the SiMT model vary at different latency. To meet the requirements, the appropriate reference should avoid forced anticipations during training and maintain high quality. Therefore, we propose to dynamically tailor reference, called *tailored reference*, for the training of SiMT model according to the latency, thereby reducing forced anticipations. We present an intuitive example of tailored reference in Figure 1. It can avoid forced predictions during training while maintaining the semantics consistent with the original sentence.

Therefore, we propose a new method for providing tailored reference to SiMT models at different latency. To accomplish this, we introduce the tailor, a shallow non-autoregressive Transformer Decoder (Gu et al., 2018), to modify ground-truth to the tailored reference. Since there is no explicit supervision to train the tailor, we quantify the requirements for the tailored reference as two reward functions and optimize them using reinforcement learning (RL). On the one hand, tailored reference should avoid forced anticipations, ensuring that the word reorderings between it and the source sentence can be handled by the SiMT model trained

| $k$ | 1 | 3 | 5 | 7 | 9 |
|---|---|---|---|---|---|
| **AR[%]** | 28.17 | 8.68 | 3.49 | 1.12 | 0.49 |

Table 1: The AR($\downarrow$) on WMT15 De$\rightarrow$En test set at different latency. $k$ belongs to Wait-$k$ policy (Ma et al., 2019) and represents the number of tokens that the target sentence lags behind the source sentence.

at that latency. To achieve this, the tailor learns from non-anticipatory reference corresponding to that latency, which can be generated by applying Wait-$k$ policy to Full-sentence model (Chen et al., 2021). On the other hand, tailored reference should maintain high quality, which can be achieved by encouraging the tailor to learn from ground-truth. Therefore, we measure the similarity between the output of tailor and both non-anticipatory reference and ground-truth, assigning them as separate rewards. The tailor can be optimized by striking a balance between these two rewards. During training, the SiMT model takes the output of the tailor as the objective and is jointly optimized with the tailor. Additionally, our method is applicable to a wide range of SiMT approaches. Experiments on three translation tasks demonstrate that our method achieves state-of-the-art performance in both fixed and adaptive policies.

## 2 Background

For a SiMT task, the model reads in the source sentence $\mathbf{x} = (x_1, ..., x_J)$ with length $J$ and generates the target sentence $\mathbf{y} = (y_1, ..., y_I)$ with length $I$ based on the policy. To describe the policy, we define the number of source tokens read in when translating $y_i$ as $g_i$. Then the policy can be represented as $\mathbf{g} = (g_1, ..., g_I)$. Therefore, the SiMT model can be trained by minimizing the cross-entropy loss:

$$\mathcal{L}_{simt} = -\sum_{i=1}^{I} \log p(y_i^\star \mid \mathbf{x}_{\leq g_i}, \mathbf{y}_{<i}), \quad (1)$$

where $y_i^\star$ represents the ground-truth token.

Our approach involves Wait-$k$ (Ma et al., 2019), HMT (Zhang and Feng, 2023b) and CTC training (Libovický and Helcl, 2018), so we briefly introduce them.

**Wait-$k$ policy**  As the most widely used fixed policy, the model reads in $k$ tokens first and then alternates writing and reading a token. It can be formalized as:

$$g_i^k = \min\{k + i - 1, J\}, \quad (2)$$

where $J$ indicates the length of the source sentence.

**HMT** Hidden Markov Transformer (HMT), which derives from the Hidden Markov Model, is the current state-of-the-art SiMT model. It treats the translation policy $\mathbf{g}$ as hidden events and the target sentence $\mathbf{y}$ as observed events. During training, HMT learns when to generate translation by minimizing the negative log-likelihood of observed events over all possible policies:

$$\mathcal{L}_{hmt} = -\log \sum_{\mathbf{g}} p(\mathbf{y} \mid \mathbf{x}, \mathbf{g}) \times p(\mathbf{g}). \quad (3)$$

**CTC** CTC (Graves et al., 2006) is applied in non-autoregressive translation (NAT) (Gu et al., 2018) due to its remarkable performance and no need for length predictor. CTC-based NAT will generate a sequence containing repetitive and blank tokens first, and then reduce it to a normal sentence based on the collapse function $\Gamma^{-1}$. During training, CTC will consider all possible sequences $\mathbf{a}$, which can be reduced to $\mathbf{y}$ using function $\Gamma^{-1}$:

$$\mathcal{L}_{ctc} = -\log \sum_{\mathbf{a} \in \Gamma(\mathbf{y})} p(\mathbf{a} \mid \mathbf{x}), \quad (4)$$

where $p(\mathbf{a} \mid \mathbf{x})$ is modeled by NAT architecture.

## 3 Method

In this section, we introduce the architecture of our model, which incorporates tailor into the SiMT model. To train the SiMT model with the tailor, we present a three-stage training method, in which the SiMT model benefits from training with tailored reference and is optimized together with the tailor. During inference, the SiMT model generates translation according to the policy. The details are introduced in the following subsections.

### 3.1 Model Architecture

We present the architecture of our method in Figure 2. Alongside the encoder and decoder, our method introduces the tailor module, which is responsible for generating a tailored reference for the SiMT model, utilizing the ground-truth as its input. Considering the efficiency of generating tailored reference, the tailor module adopts the structure of the non-autoregressive Transformer decoder (Vaswani et al., 2017). To enable the tailor to generate a tailored reference that is not limited by the length of ground-truth, it initially upsamples the ground-truth. Subsequently, it cross-attends to the output

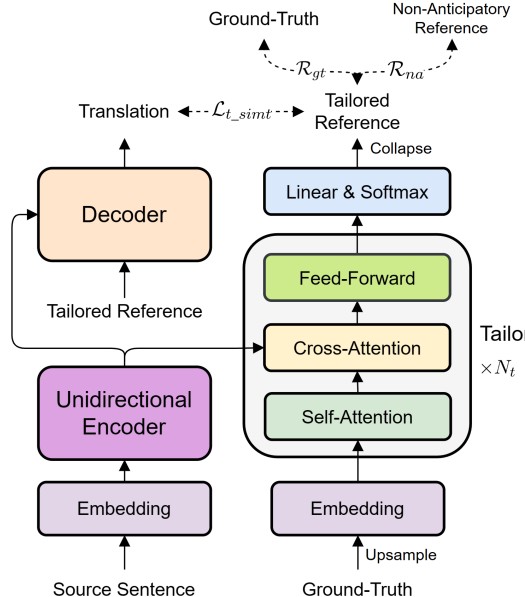

Figure 2: The architecture of our method. The tailor module modifies ground-truth to the tailored reference, which serves as the training target for the SiMT model. The tailor is induced to optimize two rewards by reinforcement learning.

of the encoder and modifies ground-truth while considering the word order of the source sentence. Finally, it transforms the output of tailor into the tailored reference by eliminating repetitive and blank tokens (Libovický and Helcl, 2018). The tailored reference replaces the ground-truth as the training objective for the SiMT model.

Given the lack of explicit supervision for training the tailor, we quantify the requirements for tailored reference into two rewards and optimize the model through reinforcement learning. We propose a three-stage training method for the SiMT model with the tailor, the details of which will be presented in the next subsection.

### 3.2 Training Method

After incorporating tailor into the SiMT model, it is essential to train the SiMT model with the assistance of tailor to get better performance. In light of this, we quantify the requirements of the tailored reference into two rewards and propose a novel three-stage training method for the training of our method. First, we train the SiMT model using ground-truth and equip the SiMT model with good translation capability. Subsequently, we use a pre-training strategy to train the tailor, enabling it to establish a favorable initial state and converge faster. Finally, we fine-tune the tailor by optimiz-

ing the two rewards using reinforcement learning, where the output of the tailor serves as the training target for the SiMT model after being reduced. In the third stage, the tailor and SiMT model are jointly optimized and share the output the of the encoder. Next, we describe our three-stage training method in detail.

**Training the Base Model** In our architecture, the tailor cross-attends to the output of the encoder to adjust ground-truth based on source information. As a result, before training the tailor module, we need a well-trained SiMT model as the base model. In our method, we choose the Wait-$k$ policy (Ma et al., 2019) and HMT model (Zhang and Feng, 2023b) as the base model for fixed policy and adaptive policy, respectively. The base model is trained using the cross-entropy loss. Once the training of the base model is completed, we optimize the tailor module, which can provide the tailored reference for the SiMT models trained across different latency settings.

**Pre-training Tailor** The tailor adopts the architecture of a non-autoregressive decoder (Gu et al., 2018). The non-autoregressive architecture has demonstrated excellent performance (Qian et al., 2020; Huang et al., 2022). Importantly, it enables the simultaneous generation of target tokens across all positions, making it highly efficient for reinforcement learning. However, if we train the tailor using reinforcement learning directly, it will converge to a suboptimal state in which the tokens at each position are some frequent tokens (Shao et al., 2022). This behavior is attributed to the exploration-based nature of reinforcement learning, highlighting the need for a favorable initial state for the model (Lopes et al., 2012). Since the tailored reference is modified from ground-truth, we let it learn from ground-truth during pre-training and then fine-tune it using reinforcement learning. The details of pre-training stage are introduced below.

To keep the output of the tailor from being limited by the length of ground-truth, the tailor upsamples ground-truth to get the input of the tailor, denoted as $\mathbf{y}'$. During training, CTC loss (Libovický and Helcl, 2018) is used to optimize the tailor.

Denoting the output of the tailor as $\mathbf{a} = (a_1, ..., a_T)$, the probability distribution modeled by the tailor can be presented as:

$$p_a(\mathbf{a} \mid \mathbf{x}, \mathbf{y}') = \prod_{t=1}^{T} p_a(a_t \mid \mathbf{x}, \mathbf{y}'), \qquad (5)$$

where $T$ is the output length of tailor and is a multiple of the length of $\mathbf{y}$. Subsequently, we can get the normal sequence $\mathbf{s}$ by applying collapse function $\Gamma^{-1}$ to $\mathbf{a}$ and the distribution of $\mathbf{s}$ is calculated by considering all possible $\mathbf{a}$:

$$p_s(\mathbf{s} \mid \mathbf{x}, \mathbf{y}') = \sum_{\mathbf{a} \in \Gamma(\mathbf{s})} p_a(\mathbf{a} \mid \mathbf{x}, \mathbf{y}'). \qquad (6)$$

To make the tailor learn from ground-truth, the tailor is optimized by minimizing the negative log-likelihood:

$$\mathcal{L}_{pt} = -\log p_s(\mathbf{y} \mid \mathbf{x}, \mathbf{y}'), \qquad (7)$$

which can be efficiently calculated through dynamic programming (Graves et al., 2006).

**RL Fine-tuning** After completing the pre-training, the tailor is already in a favorable initial state. We quantify the requirements for tailored reference as two rewards and fine-tune the tailor using reinforcement learning. We then introduce the two reward functions.

On the one hand, the tailored reference should not force the model to predict the target tokens before reading corresponding source information, which means the SiMT model can handle the word reorderings between the tailored reference and the source sentence at that latency (Zhang et al., 2022). Therefore, we make the tailor learn from non-anticipatory reference $\mathbf{y}_{na}$, which is generated by applying the corresponding Wait-$k$ policy to the Full-sentence model. It has the word order that matches the latency and maintains the original semantics (Chen et al., 2021). We employ reward $\mathcal{R}_{na}$ to measure the similarity between the output of tailor and non-anticipatory reference. On the other hand, the tailored reference should remain faithful to ground-truth. We introduce the reward $\mathcal{R}_{gt}$ to measure the similarity between ground-truth and the output of the tailor. By striking an appropriate balance between $\mathcal{R}_{na}$ and $\mathcal{R}_{gt}$, we can obtain the tailored reference.

Given the output $\mathbf{a}$ of tailor, we can obtain the normal sentence $\mathbf{s}$ by removing the repetitive and blank tokens (Libovický and Helcl, 2018). We use BLEU (Papineni et al., 2002) to measure the similarity between two sequences. Therefore, $\mathcal{R}_{na}$ and $\mathcal{R}_{gt}$ for the output of tailor is calculated as:

$$\mathcal{R}_{na}(\mathbf{s}) = \text{BLEU}(\mathbf{s}, \mathbf{y}_{na}), \qquad (8)$$

$$\mathcal{R}_{gt}(\mathbf{s}) = \text{BLEU}(\mathbf{s}, \mathbf{y}). \qquad (9)$$

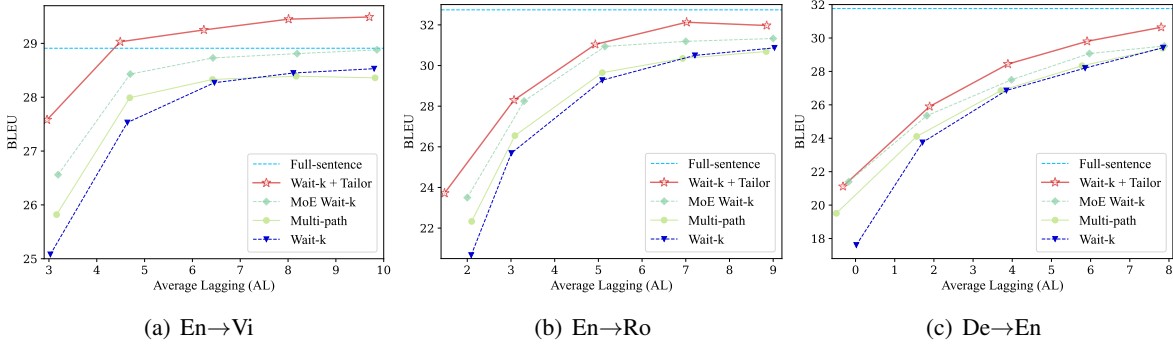

Figure 3: Translation performance of different fixed policies on En→Vi, En→Ro and De→En tasks.

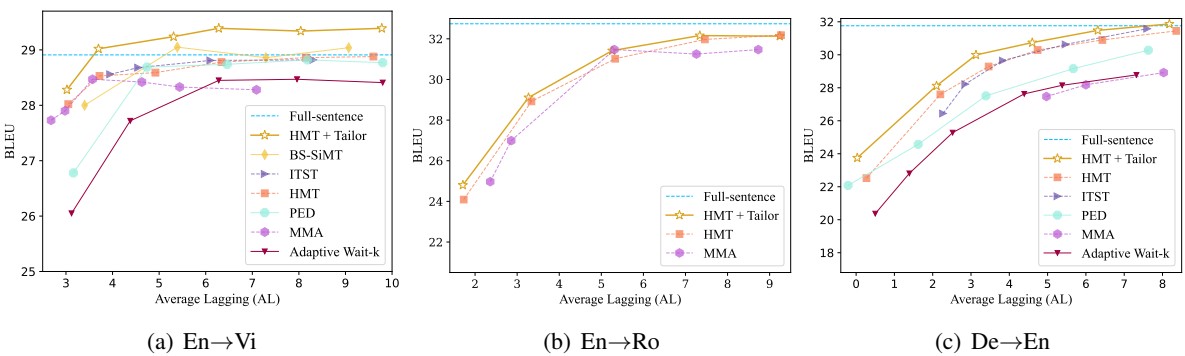

Figure 4: Translation performance of different adaptive policies on En→Vi, En→Ro and De→En tasks.

Based on these two rewards, we can obtain the final reward $\mathcal{R}$ by balancing $\mathcal{R}_{na}$ and $\mathcal{R}_{gt}$:

$$\mathcal{R}(\mathbf{s}) = (1 - \alpha)\mathcal{R}_{na}(\mathbf{s}) + \alpha\mathcal{R}_{gt}(\mathbf{s}), \qquad (10)$$

where $\alpha \in [0, 1]$ is a hyperparameter. Subsequently, we use REINFORCE algorithm (Williams, 1992) to optimize the final reward $\mathcal{R}$ to obtain the tailored reference:

$$\nabla_\theta \mathcal{J}(\theta) = \nabla_\theta \sum_{\mathbf{s}} p_s(\mathbf{s} \mid \mathbf{x}, \mathbf{y}', \theta)\mathcal{R}(\mathbf{s})$$
$$= \mathop{\mathbb{E}}_{\mathbf{s} \sim p_s} [\nabla_\theta \log p_s(\mathbf{s} \mid \mathbf{x}, \mathbf{y}', \theta)\mathcal{R}(\mathbf{s})]$$
$$= \mathop{\mathbb{E}}_{\mathbf{a} \sim p_a} [\nabla_\theta \log p_s(\Gamma^{-1}(\mathbf{a}) \mid \mathbf{x}, \mathbf{y}', \theta)\mathcal{R}(\Gamma^{-1}(\mathbf{a})].$$
$$(11)$$

where $\Gamma^{-1}$ represents the collapse function and $\theta$ denotes the parameter of tailor. During training, we sample the sequence $\mathbf{a}$ from the distribution $p_a(\mathbf{a} \mid \mathbf{x}, \mathbf{y}', \theta)$ using Monte Carlo method. As the tailor adopts a non-autoregressive structure where all positions are independent of each other, we can concurrently sample tokens for all positions from the distribution. We then apply collapse function to sequence $\mathbf{a}$ to obtain the normal sequence $\mathbf{s}$, which is used to compute the reward

$\mathcal{R}(\mathbf{s})$ and update the tailor with estimated gradient $\nabla_\theta \log p_s(\mathbf{s} \mid \mathbf{x}, \mathbf{y}', \theta)\mathcal{R}(\mathbf{s})$. In the calculation of $p_s(\mathbf{s} \mid \mathbf{x}, \mathbf{y}', \theta)$, we use dynamic programming to accelerate the process. Additionally, we adopt the baseline reward strategy to reduce the variance of the estimated gradient (Weaver and Tao, 2001).

In this stage, we utilize reinforcement learning to optimize the final reward $\mathcal{R}(\mathbf{s})$ and train the SiMT model with tailored reference using $\mathcal{L}_{t\_simt}$. As a result, the SiMT model and the tailor are jointly optimized to enhance performance.

## 4 Experiments

### 4.1 Datasets

We evaluate our method on three translation tasks.

**IWSLT15**[2] **English→Vietnamese (En→Vi)** (Cettolo et al., 2015) We use TED tst2012 as the development set and TED tst2013 as the test set. In line with Ma et al. (2020), we replace the tokens occurring less than 5 with $\langle unk \rangle$. Consequently, the vocabulary sizes of English and Vietnamese are 17K and 7.7K, respectively.

**WMT16**[3] **English→Romanian (En→Ro)** We

[2] https://nlp.stanford.edu/projects/nmt/
[3] www.statmt.org/wmt16/

use newsdev-2016 as the development set and newstest-2016 as the test set. The source and target languages employ a shared vocabulary. Other settings are consistent with Gu et al. (2018).

**WMT15[4] German→English (De→En)** Following Ma et al. (2020), we use newstest2013 as development set and newstest2015 as test set. We apply BPE (Sennrich et al., 2016) with 32K subword units and use a shared vocabulary between source and target.

## 4.2 System Settings

Our experiments involve the following methods and we briefly introduce them.

**Full-sentence** model is the conventional full-sentence machine translation model.

**Wait-$k$** policy (Ma et al., 2019) initially reads $k$ tokens and then alternates between writing and reading a source token.

**Multi-path** (Elbayad et al., 2020) introduces the unidirectional encoder and trains the model by sampling the latency $k$.

**Adaptive Wait-$k$** (Zheng et al., 2020) employs multiple Wait-$k$ models through heuristic method to achieve adaptive policy.

**MMA** (Ma et al., 2020) makes each head determine the translation policy by predicting the Bernoulli variable.

**MoE Wait-$k$** (Zhang and Feng, 2021), the current state-of-the-art fixed policy, treats each head as an expert and integrates the decisions of all experts.

**PED** (Guo et al., 2022) implements the adaptive policy via integrating post-evaluation into the fixed translation policy.

**BS-SiMT** (Guo et al., 2023) constructs the optimal policy online via binary search.

**ITST** (Zhang and Feng, 2022b) treats the translation as information transport from source to target.

**HMT** (Zhang and Feng, 2023b) models simultaneous machine translation as a Hidden Markov Model, and achieves the current state-of-the-art performance in SiMT.

**\*+100% Pseudo-Refs** (Chen et al., 2021) trains the Wait-$k$ model with ground-truth and pseudo reference, which is generated by applying Wait-$k$ policy to the Full-sentence model.

**\*+Top 40% Pseudo-Refs** (Chen et al., 2021) filters out pseudo references in the top 40% of quality to train the model with ground-truth.

**Wait-$k$ + Tailor** applies our method on Wait-$k$.

---

[4]www.statmt.org/wmt15/

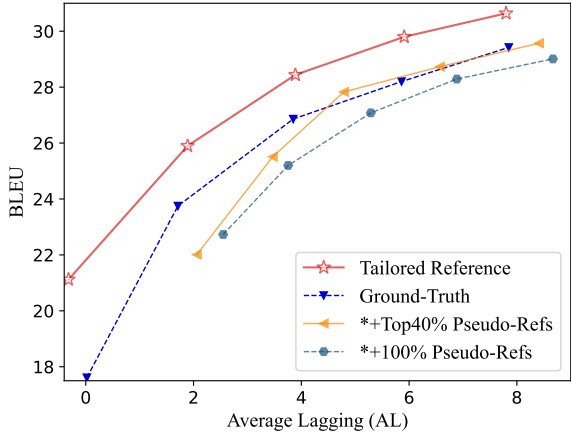

Figure 5: Translation performance of different training methods on Wait-$k$ policy.

**HMT + Tailor** applies our method on HMT.

All systems are based on Transformer architecture (Vaswani et al., 2017) and adapted from Fairseq Library (Ott et al., 2019). We apply Transformer-Small (6 layers, 4 heads) for En→Vi task and Transform-Base (6 layers, 8 heads) for En→Ro and De→En tasks. Since PED and Adaptive Wait-$k$ do not report the results on the En→Ro task, we do not compare them in the experiment. For our method, we adopt the non-regressive decoder structure with 2 layers for the tailor. We empirically set the hyperparameter $\alpha$ as 0.2. The non-anticipatory reference used for RL Fine-tuning of SiMT model is generated by Test-time Wait-$k$ method (Ma et al., 2019) with corresponding latency. Other system settings are consistent with Ma et al. (2020) and Zhang and Feng (2023b). The detailed settings are shown in Appendix C. We use greedy search during inference and evaluate all methods with translation quality estimated by BLEU (Papineni et al., 2002) and latency measured by Average Lagging (AL) (Ma et al., 2019).

## 4.3 Main Results

The performance comparison between our method and other SiMT approaches on three translation tasks is illustrated in Figure 3 and Figure 4. Our method achieves state-of-the-art translation performance in both fixed and adaptive policies. When comparing with other training methods in Figure 5, our approach also achieves superior performance.

When selecting the most commonly used Wait-$k$ policy as the base model, our method outperforms MoE Wait-$k$, which is the current state-of-the-art fixed policy. Compared to Wait-$k$ policy,

| $N_t$ | 1 | 2 | 4 |
|---|---|---|---|
| **AL** | 1.50 | **1.89** | 1.84 |
| **BLEU** | 24.43 | **25.90** | 25.46 |

Table 2: Performance of the SiMT model when the tailor has a different number of layers.

| Method | $\alpha$ | AL | BLEU |
|---|---|---|---|
| | 0.1 | 1.72 | 24.82 |
| Wait-$k$ + Tailor | 0.2 | **1.89** | **25.90** |
| | 0.3 | 1.95 | 25.30 |
| w/o Base Model | 0.2 | 1.77 | 22.89 |
| w/o Pre-training | 0.2 | 1.80 | 24.66 |
| w/o RL Fine-tuning | 0.2 | 1.86 | 24.60 |

Table 3: Ablation study on training method of the tailor and ratio between two rewards. 'w/o Base Model' removes the training stage of the base model. 'w/o Pre-training' removes the pre-training stage. 'w/o RL Fine-tuning' removes the RL fine-tuning stage.

our method brings significant improvement, especially under low latency. Wait-$k$ policy is trained on ground-truth and cannot be adjusted, which may force the model to predict tokens before reading corresponding source information (Ma et al., 2019). In contrast, our method provides a tailored reference for the SiMT model, thereby alleviating the issue of forced anticipations. Our method also exceeds Multi-path and MoE Wait-$k$. These two methods are trained using multiple Wait-$k$ policies (Elbayad et al., 2020) and gain the ability to translate under multiple latency (Zhang and Feng, 2021), but they still utilize ground-truth at all latency, leading to lower performance.

Our method can further enhance the SiMT performance by selecting adaptive policy as the base model. As the current state-of-the-art adaptive policy, HMT possesses the ability to dynamically adjust policy to balance latency and translation quality (Zhang and Feng, 2023b). However, it still relies on ground-truth for training SiMT models across different latency settings. By providing a tailored reference that matches the latency, our method can alleviate the latency burden of the SiMT model, resulting in state-of-the-art performance.

Our method also surpasses other training approaches. Ground-truth is not suitable for incremental input due to word reorderings, resulting in performance degradation (Zhang and Feng, 2022b). On the contrary, pseudo reference can avoid forced anticipations during training (Chen et al., 2021). However, it is constructed offline by applying the Wait-$k$ policy on the Full-sentence model. It imposes an upper bound on the performance of the SiMT model. The tailored reference avoids forced anticipations while maintaining high quality, leading to the best performance.

In addition to enhancing translation performance, our method effectively narrows the gap between fixed and adaptive policies. By leveraging our method, the SiMT model can achieve comparable performance to Full-sentence translation with lower latency on En→Vi and De→En tasks.

## 5 Analysis

To gain a comprehensive understanding of our method, we conducted multiple analyses. All of the following results are reported on De→En task.

### 5.1 Ablation Study

We conduct ablation studies on the structure and training method of tailor to investigate the influence of different settings. The experiments all use Wait-$k$ model as the base model with $k$ set to 3. Table 2 presents a comparison of different structures. The best performance is achieved when the tailor has 2 layers. The performance can be negatively affected by both excessive layers and insufficient layers. Table 3 illustrates the results of the ablation study on the training method. Each stage of the training method contributes to the performance of the SiMT model and the training stage of the base model has the most significant impact on the performance. This can be attributed to the fact that a well-trained encoder can provide accurate source information to the tailor, enabling the generation of appropriate tailored references. Additionally, when $\alpha$ is selected as 0.2, our method yields the best performance, indicating an optimal balance between word order and quality for the tailor.

### 5.2 Analysis of Tailored Reference

**Anticipation Rate** Furthermore, we conduct an analysis of the tailored reference to assess its influence. We first explore the rate of forced anticipations caused by using different references during training. Using the anticipation rate (AR) (Chen et al., 2021) as the metric, the results in Table 4 show that the tailored reference can effectively re-

| $k$ | 1 | 3 | 5 | 7 |
|---|---|---|---|---|
| Ground-Truth | 28.17 | 8.68 | 3.49 | 1.12 |
| Tailored Ref | 19.84 | 8.29 | 2.98 | 0.90 |

Table 4: The anticipation rate (AR[%]) when applying the Wait-$k$ policy on different references, which are based on De→En test set.

| $k$ | 1 | 3 | 5 | 7 |
|---|---|---|---|---|
| Tailored Ref | 79.67 | 86.01 | 84.83 | 92.40 |
| Non-Anti Ref | 21.87 | 24.07 | 26.29 | 27.24 |

Table 5: The quality (BLEU) of difference references compared to ground-truth for the training of Wait-$k$ policy. 'Non-Anti Ref' represents the reference generated by applying Wait-$k$ policy on Full-sentence model.

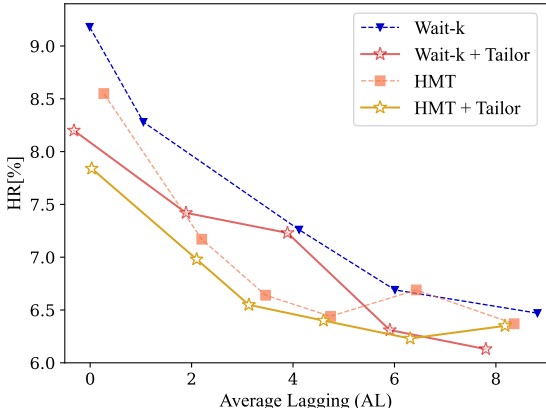

Figure 6: The hallucination rate (HR) (Chen et al., 2021) of different methods. It measures the proportion of tokens in translation that cannot find corresponding source information.

duce the forced anticipations during the training of the SiMT model under all latency. This implies that, compared to ground-truth, the word reorderings between the tailored reference and the source sentence can be more effectively handled by the SiMT model at different latency.

**Quality** However, one concern is whether the quality of tailored reference will deteriorate like non-anticipatory reference after adjusting the word order. To assess this, we compare different references with ground-truth to measure their quality. As shown in Table 5, we observe that the tailored reference exhibits significantly higher quality than the non-anticipatory reference. Therefore, our method successfully reduces the rate of forced anticipations during training while remaining faithful to ground-truth. To provide a better understanding of the tailored reference, we include several illustrative cases in Appendix B.

### 5.3 Hallucination in Translation

If the SiMT model is forced to predict target tokens before reading corresponding source information during training, there is a high likelihood of generating hallucinations during inference (Ma et al., 2019). To quantify the presence of hallucinations in the translation, we introduce hallucination rate (HR) (Chen et al., 2021) for evaluation. Figure 6 illustrates that the SiMT model trained with the tailored reference demonstrates a reduced probability of generating hallucinations. Moreover, even though the adaptive policy can adjust its behavior based on the translation status, our approach still ef-

fectively mitigates the hallucinations by alleviating the burden of latency. This signifies that minimizing forced predictions during training can enhance the faithfulness of the translation to the source sentence, thereby improving translation quality (Ma et al., 2023).

## 6 Related Work

Simultaneous machine translation (SiMT) generates translation while reading the source sentence. It requires a policy to determine the source information read when translating each target token, thus striking a balance between latency and translation quality. Current research on SiMT mainly focuses on two areas: policy improvement and adjustment of the training method.

For policy improvement, it aims to provide sufficient source information for translation while avoiding unnecessary latency. Ma et al. (2019) propose Wait-$k$ policy, which initially reads $k$ tokens and alternates between writing and reading one token. Zhang and Feng (2021) enable each head to obtain the information with a different fixed latency and integrate the decisions of multiple heads for translation. However, the fixed policy cannot be flexibly adjusted based on context, resulting in suboptimal performance. Ma et al. (2020) allow each head to determine its own policy and make all heads decide on the translation. Miao et al. (2021) propose a generative framework, which uses a re-parameterized Poisson prior to regularising the policy. Zhang and Feng (2023a) propose a segmentation policy for the source input. Zhang and Feng (2023b) model the simultaneous machine translation as a Hidden

Markov Model and achieve state-of-the-art performance. However, these methods are all trained with ground-truth, leading to forced predictions at low latency.

For the adjustment of the training method, it wants to supplement the missing full-sentence information or cater to the requirements of latency. Zhang et al. (2021) shorten the distance of source hidden states between the SiMT model and the Full-sentence model. This makes the source hidden states implicitly embed future information, but encourages data-driven prediction. On the other hand, Chen et al. (2021) try to train the model with non-anticipatory reference, which can be effectively handled by the SiMT model at that latency. However, while non-anticipatory reference can alleviate forced predictions at low latency, it hinders performance improvement at high latency.

Therefore, we want to provide a tailored reference for the SiMT models trained at different latency. The tailored reference should avoid forced anticipations and exhibit high quality. In view of the good structure and superior performance of the non-autoregressive model (Gu et al., 2018; Libovický and Helcl, 2018), we utilize it to modify the ground-truth to the tailored reference.

## 7 Conclusion

In this paper, we propose a novel method to provide a tailored reference for the training of SiMT model. Experiments and extensive analyses demonstrate that our method achieves state-of-the-art performance in both fixed and adaptive policies and effectively reduces the hallucinations in translation.

## Limitations

Regarding the system settings, we investigate the impact of the number of layers and training methods on performance. We think that further exploration of system settings could potentially yield even better results. Additionally, the tailor module aims to avoid forced anticipations and maintain faithfulness to ground-truth. If we can add language-related features to the SiMT model using a heuristic method, it may produce more suitable references for the SiMT model. We leave it for future work.

## Acknowledgment

We thank all anonymous reviewers for their valuable suggestions.

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

## A   Anticipation & Hallucination Rate

In our analyses, we evaluate the reference and translation using anticipation rate (AR) and hallucination rate (HR), respectively. We then introduce the calculation method of these two evaluation metrics in detail.

Given a sentence pair $(\mathbf{x}, \mathbf{y})$, there exists an alignment set $\mathbf{h}$, which is a set of source-target index pairs $(j, i)$ where $j^{\text{th}}$ source token $x_j$ aligns to the $i^{\text{th}}$ target token $y_i$. If we apply the policy $\mathbf{g} = (g_1, ..., g_I)$ on this sentence pair, the target token $y_i$ is forcibly anticipated ($A(i, \mathbf{h}, \mathbf{g}) = 1$) if it aligns to least one source token $x_j$ where $j > g_i$:

$$A(i, \mathbf{h}, \mathbf{g}) = \mathbb{1}[\{(j, i) \in \mathbf{h} \,|\, j > g_i\} \neq \varnothing]. \quad (12)$$

Therefore, we can define the anticipation rate (AR) of $(\mathbf{x}, \mathbf{y}, \mathbf{h})$ under the policy $\mathbf{g}$:

$$AR(\mathbf{x}, \mathbf{y}, \mathbf{h}, \mathbf{g}) = \frac{1}{|\mathbf{y}|} \sum_{i=1}^{|\mathbf{y}|} A(i, \mathbf{h}, \mathbf{g}). \quad (13)$$

The anticipation rate is a metric used to quantify the degree to which the target token is predicted before reading all relevant source tokens.

We then introduce the hallucination rate (HR). We first define the translation as $\hat{\mathbf{y}}$. A target token $\hat{y}_i$ in translation $\hat{\mathbf{y}}$ is a hallucination ($H(i, \mathbf{h})=1$) if it can not be aligned to any source token:

$$H(i, \mathbf{h}) = \mathbb{1}[\{(j, i) \in \mathbf{h}\} = \varnothing]. \quad (14)$$

Therefore, the hallucination rate can be defined as:

$$HR(\mathbf{x}, \hat{\mathbf{y}}, \mathbf{h}) = \frac{1}{|\hat{\mathbf{y}}|} \sum_{i=1}^{|\hat{\mathbf{y}}|} H(i, \mathbf{h}). \quad (15)$$

## B   Case Study

We also provide two cases in the De→En test set to understand our method. The cases are shown in Figure 7 and Figure 8. It presents that the tailored reference is more consistent with the word order requirements of the specific latency.

In Figure 7, if we train the SiMT model on Wait-1 policy with the ground-truth, it will be forced to predict 'with him' before reading 'mit ihm' during training. However, training the SiMT model with tailored reference will eliminate forced predictions by adjusting 'like that' to 'such' and positioning it forward. Importantly, the tailored reference also maintains the original semantics. This shows that

| Source Sentence | ich | hatte | nie | ein | **solches** | Gespräch | **mit** | **ihm** | . |
|---|---|---|---|---|---|---|---|---|---|
| | (I) | (had) | (never) | (a) | (such) | (conversation) | (with) | (him) | |
| Ground-Truth | I | never | had | a | conversation | **with** | **him** | **like** | **that** | . |
| Tailored Ref | I | had | never | had | **such** | a | conversation | **with** | **him** | . |

Figure 7: Case study of #319 in De→En test set. The tokens marked in the same color share the same semantics. The tailored reference is more suitable when the model is trained with Wait-1 policy.

| Source Sentence | und | vor | **einer** | **Online** | @-@ | **Bestellung** | sollte | man | prüfen | , | so |
|---|---|---|---|---|---|---|---|---|---|---|---|
| | (and) | (before) | (an) | (online) | | (order) | (should) | (you) | (check) | | (so) |
| | Gault | , | ob | das | Buch | tatsächlich | vorrätig | ist . | | | |
| | (Gault) | (if) | (the) | (book) | (actually) | (in stock) | (is) | | | | |
| Ground-Truth | before **ordering** **online** , Gault says , be sure to check if the book is actually in stock . |
| Tailored Ref | and before **an online order** , Gault says , be sure to check if the book is actually in stock . |

Figure 8: Case study of #1869 in De→En test set. The tokens marked in the same color share the same semantics. The tailored reference is more suitable when the model is trained with Wait-1 policy.

the order of the source sentence and the tailored sentence is consistent, which makes it suitable for Wait-1 policy.

In Figure 8, using ground-truth as the training target of Wait-1 policy also forces the model to predict 'ordering online' before reading 'Online' and 'Bestellung'. In contrast, by replacing 'ordering online' with 'an online order', the word order of tailored reference is the same as the source sentence, thereby avoiding forced anticipations during the training of Wait-1 policy.

## C Hyperparameters

The system settings on three translation tasks are shown in Table 6. For more detailed implementation issues, please refer to our publicly available code.

## D Numerical Results

In addition to the translation performance comparison in Figure 3 and Figure 4, we also provide corresponding numerical results for reference. Table 7, 8, 9 respectively report the numerical results on IWSLT15 En→Vi, WMT16 En→Ro and WMT15 De→En measured by AL (Ma et al., 2019) and BLEU (Papineni et al., 2002).

| Hyperparameter | IWSLT15 En→Vi | WMT16 En→Ro | WMT15 De→En |
|---|---|---|---|
| encoder layers | 6 | 6 | 6 |
| encoder attention heads | 4 | 8 | 8 |
| encoder embed dim | 512 | 512 | 512 |
| encoder ffn embed dim | 1024 | 2048 | 2048 |
| decoder layers | 6 | 6 | 6 |
| decoder attention heads | 4 | 8 | 8 |
| decoder embed dim | 512 | 512 | 512 |
| decoder ffn embed dim | 1024 | 2048 | 2048 |
| tailor layers | 2 | 2 | 2 |
| tailor attention heads | 8 | 8 | 8 |
| tailor embed dim | 512 | 512 | 512 |
| tailor ffn embed dim | 2048 | 2048 | 2048 |
| dropout | 0.1 | 0.3 | 0.3 |
| optimizer | adam | adam | adam |
| adam-$\beta$ | (0.9, 0.98) | (0.9, 0.98) | (0.9, 0.98) |
| clip-norm | 0 | 0 | 0 |
| lr | 5e-4 | 5e-4 | 5e-4 |
| lr scheduler | inverse sqrt | inverse sqrt | inverse sqrt |
| warmup-updates | 4000 | 4000 | 4000 |
| warmup-init-lr | 1e-7 | 1e-7 | 1e-7 |
| weight decay | 0.0001 | 0.0001 | 0.0001 |
| label-smoothing | 0.1 | 0.1 | 0.1 |
| max tokens | 16000 | 8192×4 | 8192×4 |

Table 6: Hyperparameters of our experiments.

## IWSLT15 En→Vi

### Full-sentence

| | AL | BLEU |
|---|---|---|
| | 22.41 | 28.8 |

### Wait-$k$

| $k$ | AL | BLEU |
|---|---|---|
| 1 | 3.03 | 25.28 |
| 3 | 4.64 | 27.53 |
| 5 | 6.46 | 28.27 |
| 7 | 8.11 | 28.45 |
| 9 | 9.80 | 28.53 |

### MoE Wait-$k$

| $k$ | AL | BLEU |
|---|---|---|
| 1 | 3.19 | 26.56 |
| 3 | 4.70 | 28.43 |
| 5 | 6.43 | 28.73 |
| 7 | 8.19 | 28.81 |
| 9 | 9.86 | 28.88 |

### Wait-$k$ + Tailor

| $k$ | AL | BLEU |
|---|---|---|
| 1 | 2.96 | 27.58 |
| 3 | 4.49 | 29.03 |
| 5 | 6.24 | 29.25 |
| 7 | 8.01 | 29.45 |
| 9 | 9.70 | 29.49 |

### HMT

| $L, K$ | AL | BLEU |
|---|---|---|
| 1, 2 | 3.05 | 28.02 |
| 2, 2 | 3.72 | 28.53 |
| 4, 2 | 4.92 | 28.59 |
| 5, 4 | 6.34 | 28.78 |
| 6, 4 | 8.15 | 28.86 |
| 7, 6 | 9.60 | 28.88 |

### HMT + Tailor

| $L, K$ | AL | BLEU |
|---|---|---|
| 1, 2 | 3.02 | 28.28 |
| 2, 2 | 3.70 | 29.02 |
| 4, 2 | 5.31 | 29.24 |
| 5, 4 | 6.28 | 29.39 |
| 6, 4 | 8.04 | 29.34 |
| 7, 6 | 9.78 | 29.39 |

Table 7: Numerical results of IWSLT15 En→Vi.

## WMT16 En→Ro

### Full-sentence

| | AL | BLEU |
|---|---|---|
| | n/a | 32.74 |

### Wait-$k$

| $k$ | AL | BLEU |
|---|---|---|
| 1 | 2.09 | 20.67 |
| 3 | 3.01 | 25.69 |
| 5 | 5.09 | 29.28 |
| 7 | 7.21 | 30.50 |
| 9 | 9.03 | 30.87 |

### MoE Wait-$k$

| $k$ | AL | BLEU |
|---|---|---|
| 1 | 2.00 | 23.50 |
| 3 | 3.30 | 28.25 |
| 5 | 5.15 | 30.94 |
| 7 | 7.00 | 31.19 |
| 9 | 9.00 | 31.33 |

### Wait-$k$ + Tailor

| $k$ | AL | BLEU |
|---|---|---|
| 1 | 1.48 | 23.72 |
| 3 | 3.07 | 28.30 |
| 5 | 4.93 | 31.04 |
| 7 | 7.02 | 32.13 |
| 9 | 8.85 | 31.97 |

### HMT

| $L, K$ | AL | BLEU |
|---|---|---|
| 2, 4 | 1.73 | 24.09 |
| 3, 6 | 3.34 | 28.92 |
| 5, 6 | 5.33 | 31.02 |
| 7, 6 | 7.46 | 31.97 |
| 9, 8 | 9.27 | 32.18 |

### HMT + Tailor

| $L, K$ | AL | BLEU |
|---|---|---|
| 2, 4 | 1.71 | 24.80 |
| 3, 6 | 3.27 | 29.11 |
| 5, 6 | 5.26 | 31.42 |
| 7, 6 | 7.34 | 32.16 |
| 9, 8 | 9.24 | 32.13 |

Table 8: Numerical results of WMT16 En→Ro.

| WMT16 En→Ro | | |
|:---:|:---:|:---:|
| ***Full-sentence*** | | |
| | AL | BLEU |
| | n/a | 31.76 |
| ***Wait-$k$*** | | |
| $k$ | AL | BLEU |
| 1 | 0.02 | 17.61 |
| 3 | 1.71 | 23.75 |
| 5 | 3.85 | 26.86 |
| 7 | 5.86 | 28.20 |
| 9 | 7.85 | 29.42 |
| ***MoE Wait-$k$*** | | |
| $k$ | AL | BLEU |
| 1 | -0.18 | 21.40 |
| 3 | 1.82 | 25.35 |
| 5 | 3.98 | 27.50 |
| 7 | 5.97 | 29.07 |
| 9 | 7.88 | 29.52 |
| ***Wait-$k$ + Tailor*** | | |
| $k$ | AL | BLEU |
| 1 | -0.32 | 21.12 |
| 3 | 1.89 | 25.90 |
| 5 | 3.89 | 28.44 |
| 7 | 5.91 | 29.80 |
| 9 | 7.80 | 30.64 |
| ***HMT*** | | |
| $L, K$ | AL | BLEU |
| $-1, 4$ | 0.27 | 22.52 |
| $2, 4$ | 2.20 | 27.60 |
| $3, 6$ | 3.46 | 29.29 |
| $5, 6$ | 4.74 | 30.29 |
| $7, 6$ | 6.43 | 30.90 |
| $9, 8$ | 8.36 | 31.45 |
| ***HMT + Tailor*** | | |
| $L, K$ | AL | BLEU |
| $-1, 4$ | 0.03 | 23.75 |
| $2, 4$ | 2.10 | 28.12 |
| $3, 6$ | 3.13 | 29.99 |
| $5, 6$ | 4.60 | 30.74 |
| $7, 6$ | 6.31 | 31.48 |
| $9, 8$ | 8.18 | 31.87 |

Table 9: Numerical results of WMT15 De→En.