# OpenReview forum: "Simultaneous Machine Translation with Tailored Reference"
_EMNLP/2023/Conference — EMNLP 2023 Findings_

### Official Review · Reviewer_DPEY · 2023-08-05

**Soundness:** 4

**Excitement:**

4: Strong: This paper deepens the understanding of some phenomenon or lowers the barriers to an existing research direction.

**Paper Topic And Main Contributions:**

This paper is about avoiding forced anticipations during training a simultaneous machine translation (SiMT) model. Based on the conventional training of SiMT, this paper proposes to add a decoder (tailor) module which is trained on a combined target of the non-anticipatory reference and the ground truth. In this way, the tailor module could act as a regularizer to avoid forced anticipations.

The main contributions of the paper are:
1. The motivation is clear, and the resolved issue is also practical for SiMT;
2. Extensive experiments are conducted to support the main claims, and the results are convincing.
3. Thorough analysis and ablation study.

**Questions For The Authors:**

1. In Figures 3 and 4 for the translation direction of En->Vi, the proposed method even outperforms the full-sentence translation that I consider as the upper bound. Do you have any explanation for this? I think one reason might be the first point in reasons to reject. You use the tailor to implicitly generate some synthetic data.

**Reasons To Accept:**

1. The motivation is clear, and the resolved issue is also practical for SiMT;
2. Extensive experiments are conducted to support the main claims, and the results are convincing.
3. Thorough analysis and ablation study.

**Reasons To Reject:**

1. Missing baseline: the proposed method is very related to the target-side augmentation method. It's better to add some related baselines, like forward-translation, i.e. applying a trained SiMT model to translate the source sentence and combining these synthetic data with the original data together to further trains the SiMT model.
2. The wring should be improved, especially the sections about the proposed method.
3. Experimental details are missing, like batch size, learning rater and so on, which makes it difficult to reproduce.

**Reproducibility:**

3: Could reproduce the results with some difficulty. The settings of parameters are underspecified or subjectively determined; the training/evaluation data are not widely available.

**Reviewer Confidence:**

4: Quite sure. I tried to check the important points carefully. It's unlikely, though conceivable, that I missed something that should affect my ratings.

---

> ### Author Rebuttal · Authors · 2023-08-27
>
> Thanks for your valuable and insightful comments.
>
> &ensp;
>
> Q1: Append the results of the additional baseline?\
> A1: Thank you for your valuable suggestion. We take your feedback into consideration and include an additional baseline. It involves optimizing the SiMT model with ground-truth and non-anticipatory reference concurrently. The comparison of our method with other baselines is shown in the table below.
>
> | Reference | AL | BLEU |
> | :-----| :----: | :----: |
> | **Tailored Reference** | **1.89** | **25.90** |
> | Non-anticipatory Reference | 2.55 | 22.73 |
> | Ground-Trurh, Non-anticipatory Reference | 3.39 | 24.23 |
>
> &ensp;&ensp;&ensp; As evident from the table, our method exhibits better latency-quality trade-offs compared to the other baselines, demonstrating the effectiveness of our method. We sincerely appreciate your suggestion and will ensure that the newly added baseline results are included in our paper.
>
> &ensp;
>
> Q2: Lack of some training details?\
> A2: We deeply appreciate your attention to detail and your valuable feedback. Due to space limitations, we mention in Section 4.2 that experimental details not listed are kept consistent with those of [1]. The experimental settings are presented in the following table:
>
> | Hyperparameter | WMT15 De->En | WMT16 En->Ro | IWSLT15 En->Vi |
> | :-----| :----: | :----: | :----: |
> | model | Transformer Base | Transformer Base | Transformer Small |
> | dropout | 0.3 | 0.3 | 0.1 |
> | optimizer | adam | adam | adam |
> | adam-$\beta$ | (0.9, 0.98) | (0.9, 0.98) | (0.9, 0.98) |
> | lr | 5e-4 | 5e-4 | 5e-4 |
> | weight decay | 1e-4 | 1e-4 | 1e-4 |
> | max tokens | 8192$\times$4 | 8192$\times$4 | 16000 |
>
> &ensp;&ensp;&ensp; We acknowledge the significance of these details for replicability and will ensure to provide a more comprehensive account of our experimental settings in the next version. Additionally, to enhance the reproducibility of our work, we will make our code publicly available.
>
> [1] Ma et al., 2020. Monotonic multihead attention. In ICLR2020.
>
> &ensp;
>
> Q3: Do you have any explanation for the phenomenon that the proposed method even outperforms the full-sentence translation in the direction of En->Vi?\
> A3: Previous approaches such as HMT have demonstrated that the SiMT model can outperform the performance of the full-sentence translation under high latency. I think there are mainly two reasons why our method outperforms the full-sentence translation. The first interpretation is indeed what you mentioned: "**the tailor implicitly generates some synthetic data**," which can help the model reduce hallucinations and generate more reliable translations. Another interpretation is that our method empowers the adaptive policy to acquire more accurate policies. This enables the translation model to **disregard irrelevant source tokens and concentrate on the necessary source tokens**, thereby obtaining high-quality translation.
>
> &ensp;
>
> If our answers can solve your questions, we would appreciate it if you could increase the score.

---

### Official Review · Reviewer_XdDD · 2023-08-10

**Soundness:** 4

**Excitement:**

4: Strong: This paper deepens the understanding of some phenomenon or lowers the barriers to an existing research direction.

**Paper Topic And Main Contributions:**

This paper addresses the problem of finding optimal training references for simultaneous machine translation (SiMT) models under varying latency constraints. The key insight is that using the full ground truth translation as the training target is suboptimal, as it forces premature anticipations at low latency and degrades quality at high latency.



The paper proposes to dynamically generate tailored training references by modifying the ground truth. This is achieved via a trainable "tailor" model, which is trained based on NAT pretraining and RL training.

**Reasons To Accept:**

The positive results demonstrate that adapting the ground-truth translation with a third model can indeed create a better reference for training SIMT models.

**Reasons To Reject:**

I'm concerned that the proposed method is overly complicated. Since we are creating non-anticipatory references by applying the wait-K policy to the full-sentence model. Looking at Figure 2, I think there are some additional baselines we can construct easily that do not require training a "tailor" model:


1/ Directly training the SIMT model using non-anticipatory reference

2/ Training the SIMT model with a mixed loss

     alpha * L_ground_truth + (1 - alpha) * L_non_anticipatory

Both are related to mode distillation.

**Reproducibility:**

3: Could reproduce the results with some difficulty. The settings of parameters are underspecified or subjectively determined; the training/evaluation data are not widely available.

**Reviewer Confidence:**

4: Quite sure. I tried to check the important points carefully. It's unlikely, though conceivable, that I missed something that should affect my ratings.

---

> ### Author Rebuttal · Authors · 2023-08-27
>
> Thanks for your valuable and insightful comments.
>
> &ensp;
>
> Q1: Concern that our method is overly complicated?\
> A1: The biggest difference between our method and previous methods is the online generation of tailored references. This online generation method ensures that the tailored references not only consider latency requirements to avoid forced predictions, but also maintain the high quality of the reference. The experiments in Figure 5 show that **our online method is remarkably more effective than previous offline methods**.\
> &ensp;&ensp;&ensp; Regarding the training complexity, our method introduces two additional training processes: pre-training of the tailor and RL fine-tuning. However, these two training processes require only five epochs and seven epochs respectively to converge. Therefore, **the associated training costs for the two processes are relatively modest**.\
> &ensp;&ensp;&ensp; Considering the substantial improvement in performance, the moderate increase in complexity introduced by our method is acceptable.
>
> &ensp;
>
> Q2: Append the results of some additional baselines?\
> A2: Thank you for suggesting additional baselines for our evaluation. We greatly appreciate your insights.\
> &ensp;&ensp;&ensp; Regarding directly training the SiMT model using non-anticipatory reference, it has been explored by [1]. We implement this method on the WMT15 De->En dataset and the results are in Figure 5. We also implement the second baseline you propose. The corresponding results are included in the following table for comparison.
>
> | Method | AL | BLEU |
> | :-----| :----: | :----: |
> | **Tailored Reference** | **1.89** | **25.90** |
> | Non-anticipatory Reference | 2.55 | 22.73 |
> | Mixed Loss | 3.39 | 24.23 |
>
> &ensp;&ensp;&ensp; It can be seen that our method significantly outperforms other baselines and achieves better latency-quality trade-offs. We will include the above results in the paper to demonstrate the effectiveness of our proposed method.
>
> [1] Chen et al., 2021. Improving simultaneous translation by incorporating pseudo-references with fewer reorderings. In EMNLP2021.
>
> &ensp;
>
> If our answers can solve your questions, we would appreciate it if you could increase the score.

---

### Official Review · Reviewer_aHPk · 2023-08-11

**Soundness:** 4

**Excitement:**

4: Strong: This paper deepens the understanding of some phenomenon or lowers the barriers to an existing research direction.

**Paper Topic And Main Contributions:**

This paper presents a novel approach to synthesizing high-quality pseudo-references tailored to a given latency by employing an additional Non-Autoregressive Translation (NAT) module. The authors use Reinforcement Learning to jointly optimize the translation model and the NAT module, resulting in clear advantages in the BLEU-AL curves across different datasets and policies. Overall, this work shows promise. However, some descriptions in the paper could be clearer, such as the method for upsampling the ground-truth and estimating the anticipation rate.

**Questions For The Authors:**

1. How do the authors upsample the ground-truth for the NAT module?
2. How do the authors estimate the anticipation rate?

**Reasons To Accept:**

1. A novel and effective approach.

**Reasons To Reject:**

1. Some descriptions are not clear.

**Reproducibility:**

3: Could reproduce the results with some difficulty. The settings of parameters are underspecified or subjectively determined; the training/evaluation data are not widely available.

**Reviewer Confidence:**

2: Willing to defend my evaluation, but it is fairly likely that I missed some details, didn't understand some central points, or can't be sure about the novelty of the work.

---

> ### Author Rebuttal · Authors · 2023-08-27
>
> Thanks for your valuable and insightful comments.
>
> &ensp;
>
> Q1: How do the authors upsample the ground-truth for the NAT module?\
> A1: In our experimental setup, we upsample ground-truth by duplicating each target token twice.
>
> &ensp;
>
> Q2: How do the authors estimate the anticipation rate?\
> A2: The anticipation rate measures the proportion of forced predictions that arise when a policy and parallel sentence are provided.  In practical calculations, we first obtain alignments from the parallel sentence. For each target token, if there are still relevant source tokens that have not been read in while translating that target token, it is considered a forced prediction[1]. A comprehensive calculation method is elaborated in detail in Appendix A for your reference.
>
> [1] Chen et al., 2021. Improving simultaneous translation by incorporating pseudo-references with fewer reorderings. In EMNLP2021.
>
> &ensp;
>
> If our answers can solve your questions, we would appreciate it if you could increase the score.

---

### Official Review · Reviewer_Boo4 · 2023-08-11

**Soundness:** 3

**Excitement:**

3: Ambivalent: It has merits (e.g., it reports state-of-the-art results, the idea is nice), but there are key weaknesses (e.g., it describes incremental work), and it can significantly benefit from another round of revision. However, I won't object to accepting it if my co-reviewers champion it.

**Paper Topic And Main Contributions:**

This paper proposes to provide a tailored reference for the SiMT models trained at different latency varies. The tailored reference aims to avoid forced anticipations and maintain faithfulness to ground-truth. The paper investigates the impact of the number of layers and training methods on performance and suggests that further exploration of system settings could yield even better results. The main contribution of this paper is that it improves the performance of simultaneous machine translation by incorporating pseudo-references with fewer reorderings.

**Questions For The Authors:**

a. The motivation in this paper is not very clear. It is suggested that the author give examples to explain what kind of problems in simultaneous machine translation have been solved, and the proportion and importance of these problems in this area.
b. It is suggested that the author supplement the pipeline details of the three-step method and demonstrate their necessity through experiments.
c. As we all know, hallucinations are very serious in simultaneous machine translation. Can some practical examples be provided to illustrate how the method proposed in this paper alleviates this problem?

**Reasons To Accept:**

The strengths of this paper are that it proposes a tailored reference for SiMT models trained at different latencies, which achieves superior performance. It also investigates the impact of the number of layers and training methods on performance and suggests that further exploration of system settings could yield even better results. The paper also incorporates pseudo-references with fewer reorderings to improve simultaneous machine translation, which can inform future research.

**Reasons To Reject:**

The definition of motivation and tailor in this paper are not very clear, and the author does not clearly introduce what kind of problems are to be solved in simultaneous machine translation. From the only examples available, it can be seen that the paper designed a tailored reference and adjusted the word order for joint training. However, whether this would affect the fluency of the translation was not discussed, and how much of the ability to adjust the order was lost. This paper proposes a three-stage training method for training the SiMT model with the tailor, but the details of the pipeline were not very clear, and the ablation experiment did not demonstrate the necessity and effectiveness of these three stages.

**Reproducibility:**

3: Could reproduce the results with some difficulty. The settings of parameters are underspecified or subjectively determined; the training/evaluation data are not widely available.

**Reviewer Confidence:**

4: Quite sure. I tried to check the important points carefully. It's unlikely, though conceivable, that I missed something that should affect my ratings.

**Typos Grammar Style And Presentation Improvements:**

different latency -> different latencies
the Full-sentence -> Full-sentence
provides tailored reference -> provides a tailored reference
tailored reference -> the tailored reference

---

> ### Author Rebuttal · Authors · 2023-08-27
>
> Thanks for your valuable and insightful comments.
>
> &ensp;
>
> Q1: Give more examples to explain what kind of problems in SiMT have been solved, and the proportion and importance of these problems in this area?\
> A1: We apologize for causing confusion. The SiMT model performs translation while reading the source sentence. Previous SiMT methods are trained using standard full-sentence corpora, where reordering between source and target sentences occurs. **This leads to forced predictions during training and results in hallucinations during inference, posing a significant challenge for SiMT models**. As illustrated in Figure 1, the wait-*1* model will be forced to predict "an enjoyable activity" before reading in the aligned source content if it is trained with ground-truth. The problem of forced prediction is pervasive in the SiMT and progressively worsens as the latency decreases, as evident in Table 1.\
> &ensp;&ensp;&ensp; Therefore, our method is designed to dynamically provide tailored references to the SiMT model, enhancing its translation performance under all latency while avoiding forced predictions. The experimental results in Figure 5 demonstrate that our method is capable of significantly reducing hallucinations in both fixed and adaptive policies.\
> &ensp;&ensp;&ensp; We greatly appreciate your inquiry, and we will underscore the specific challenges faced by SiMT models and the problems our method addresses in the next version.
>
> &ensp;
>
> Q2: Supplement the pipeline details of the three-step method and demonstrate their necessity through experiments?\
> A2: I apologize for any confusion that may have arisen. Due to space limitations, we mention in Section 4.2 that experimental details not listed are kept consistent with those of [1]. The experimental settings are presented in the following table.
>
> | Hyperparameter | WMT15 De->En | WMT16 En->Ro | IWSLT15 En->Vi |
> | :-----| :----: | :----: | :----: |
> | model | Transformer Base | Transformer Base | Transformer Small |
> | dropout | 0.3 | 0.3 | 0.1 |
> | optimizer | adam | adam | adam |
> | adam-$\beta$ | (0.9, 0.98) | (0.9, 0.98) | (0.9, 0.98) |
> | lr | 5e-4 | 5e-4 | 5e-4 |
> | weight decay | 1e-4 | 1e-4 | 1e-4 |
> | max tokens | 8192$\times$4 | 8192$\times$4 | 16000 |
>
> &ensp;&ensp;&ensp; We will include these experimental settings in the next version and make our code publicly available to enhance reproducibility.\
> &ensp;&ensp;&ensp; We demonstrate the necessity of pre-training of the tailor in Table 3 and supplement additional experiments to illustrate the necessity of the other two training stages, as illustrated in the table below.
>
> | Method | AL | BLEU |
> | :----------------------------------| :-------------: | :----------: |
> | **Wait-k + Tailor** | **1.89** | **25.90** |
> | $~~$ w/o Base Model | 1.77 | 22.89 |
> | $~~$ w/o Pre-training | 1.80 | 24.66 |
> | $~~$ w/o RL Fine-tuning| 1.86 | 24.60 |
>
> &ensp;&ensp;&ensp; From the above results, it is evident that the three-step training approach achieves the best performance, thereby showcasing the importance of each training stage.\
> &ensp;&ensp;&ensp; In line with your valuable suggestions, we will incorporate the pipeline details and the aforementioned experimental results into the paper.
>
> [1] Ma et al., 2020. Monotonic multihead attention. In ICLR2020.
>
> &ensp;
>
> Q3: Provide some practical examples to illustrate how the method proposed in this paper alleviates hallucinations?\
> A3: I apologize for any confusion. We provide statistical results in Figure 6 to demonstrate that our method can effectively alleviate hallucinations in both fixed and adaptive policies under different latency. In order to provide a practical illustration of how our proposed method alleviates hallucinations, we present the case of #366 in De->En test set.
>
> | Sentence | Example |
> | :-----------------------------------------| :---------------------------------------------------------------- |
> | Source Sentence | Auf Shakespeare war ich regelrecht allergisch . |
> | Ground-Truth | I had a real allergy to Shakespeare . |
> | Tailored Reference| on Shakespeare , I was quite allergic . |
> | Translation (Wait-1) | on Shakespeare , I was quite right to say that . |
> | Translation (Wait-1 + Tailor) | on Shakespeare , I was quite allergic . |
>
> &ensp;&ensp;&ensp; The example illustrates that our method generates a tailored reference that adheres closely to the word order of the source sentence. When the SiMT model is trained with the tailored reference, it can avoid forced predictions during training, leading to a reduction in hallucinations during inference.\
> &ensp;&ensp;&ensp; We appreciate your suggestion and will certainly include this example in the paper to provide a clearer understanding of the practical impact of our proposed method.
>
> &ensp;
>
> Q4: About typos grammar style and presentation Improvements?\
> A4: Thank you for your meticulous review and for providing valuable suggestions for improvement. We will make the necessary revisions and refinements in the next version to enhance the clarity, grammar, style, and overall presentation of the paper. Your feedback is greatly appreciated.
>
> &ensp;
>
> If our answers solve your problem and eliminate your concerns, we would appreciate it if you could increase the score.

---

### Official Review · Reviewer_AVCF · 2023-08-11

**Soundness:** 2

**Excitement:**

2: Mediocre: This paper makes marginal contributions (vs non-contemporaneous work), so I would rather not see it in the conference.

**Paper Topic And Main Contributions:**

This paper works on improving the simultaneous translation task, where the authors take the direction of generating pseudo references. Specifically, the authors proposed a CTC-based pseudo-reference generator. The generator aims to generate pseudo references that are close to the groundtruth and of low anticipation rate. The training of the generator is through RL. The proposed method can be combined with existing approaches, and the authors show that it can improve the wait-k method and the HMM Transformer.

**Questions For The Authors:**

1. Why is 1.89 highlighted in Table 3?
2. What is Nt in Figure 2?

**Reasons To Accept:**

1. Generating pseudo references for simultaneous translation is a reasonable direction.
2. Using RL to train the pseudo-reference generator is somewhat novel, but there needs more details for reproducibility.

**Reasons To Reject:**

1. The motivation for the model design is not clear. Specifically, the authors should explain the following:
      1. Why use non-autoregressive models to get pseudo reference?
      2. Why share the encoder?
      3. Why joint training? It does not seem like the SiMT model can help the pseudo-reference generator. The direct approach should be training the reference generator first, then training the SiMT model.
2. Not comparing with appropriate correct baseline models. Admittedly, the proposed approach is orthogonal to HMM Transformer and wait-K, which means the improvements can stack. However, the proposed method should be compared with similar methods that generate pseudo references, such as [Chen+ EMNLP21] and [Zhang&Feng EMNLP22].
3. Equation (5-7) is not clear. What is p_a, p_s? Why is y in this condition? Particularly, Eq. 7 computes p(y|x,y), which does not make sense. L273 "Efficient calculated through CTC" is not clear, and it should be "through dynamic programming".
4. There is a lack of details for replication, e.g., model size and training/finetuning schedule.
5.  L67-L70 appears to be circular reasoning.
6. Appendix A highly resembles the content of Section 4 of [Chen+ EMNLP21], which should be omitted.
7. The tables need to be clear. Which datasets are they based on?

**Reproducibility:**

2: Would be hard pressed to reproduce the results. The contribution depends on data that are simply not available outside the author's institution or consortium; not enough details are provided.

**Reviewer Confidence:**

4: Quite sure. I tried to check the important points carefully. It's unlikely, though conceivable, that I missed something that should affect my ratings.

---

> ### Author Rebuttal · Authors · 2023-08-27
>
> Thank you for your thorough and valuable evaluation.
>
> &ensp;
>
> Q1: Why use non-autoregressive models to get pseudo reference?\
> A1: There are two main reasons for employing non-autoregressive models to generate tailored references.
> 1. Our method aims to provide tailored references for the SiMT model online, ensuring translation quality while minimizing forced predictions. To generate tailored references that meet these requirements, we leverage reinforcement learning to optimize the tailor.  In order to accommodate the exploration-based nature of reinforcement learning, we choose the non-autoregressive structure, **enabling independent sampling of tokens across all positions**. This design facilitates concurrent token sampling, leading to significant reductions in training costs.
> 2. Non-autoregressive translation models have demonstrated **comparable performance to autoregressive models**, showcasing their capability to generate high-quality references.
>
> &ensp;&ensp;&ensp; Hence, we select the non-autoregressive model as the mechanism for generating tailored references. We appreciate your reminder and will emphasize this aspect in the paper.
>
> &ensp;
>
> Q2: Why share the encoder?\
> A2: Our method involves adapting the ground-truth to a tailored reference, aligning it more closely with the source input of the SiMT model. Consequently, the tailor needs to obtain the representation of the source sentence. By sharing the encoder, the tailor can take into account the input attributes of the SiMT model, resulting in the generation of a tailored reference that is well-suited for the SiMT model. 	Your question is appreciated, and we will provide additional explanations on sharing the encoder.
>
> &ensp;
>
> Q3: Why joint training?\
> A3: The primary distinction of our method from prior approaches lies in the online generation of tailored references. This dynamic reference generation approach, grounded in source representation, is more suitable as the training objective of the SiMT model. **To match the dynamic reference generation method, we employ the shared encoder and joint training**.\
> &ensp;&ensp;&ensp; In scenarios where the encoder is shared, if the approach of training the tailor first and then the SiMT model is adopted, it does not yield optimal performance. Because **training the SiMT model separately can induce alterations in the parameters of the encoder, rendering the tailor incompatible with the modification of the encoder**. Consequently, the quality of the generated tailored reference could degrade, which is not suitable for the SiMT model.\
> &ensp;&ensp;&ensp; We appreciate your insights, and we will incorporate an explanation of joint training for further clarity.
>
> &ensp;
>
> Q4: Not comparing with other methods that generate pseudo references?\
> A4: We apologize for any misunderstanding that I may have caused. We compare our method with another training method [1] on De->En task in section 5.2. This comparison demonstrates our method outperforms previous training methods by avoiding forced anticipations while maintaining high quality.  In line with your suggestion, we will move the experimental results from this section to the main experiment section for better clarity.
>
> [1] Chen et al., 2021. Improving simultaneous translation by incorporating pseudo-references with fewer reorderings. In EMNLP2021.
>
> &ensp;
>
> Q5: What is p_a, p_s? Why is y in the condition of Equation (5-7)?  What is the meaning of Eq. 7? "Efficient calculated through CTC" is not clear?\
> A5: **p_a** refers to the probability distribution of the output sequence **a** generated by the tailor. The length of **a** corresponds to the input length of the tailor and may comprise repetitive and blank tokens. On the other hand, **p_s** represents the probability distribution of the normal sentence **s**. It is calculated by considering all possible sequences **a** that can be reduced to **s**.\
> &ensp;&ensp;&ensp; The rationale for employing **y** as a condition in Equation (5-7) stems from the fact that the tailor generates the tailored reference based on the ground-truth. As illustrated in Figure 2, the tailor unsamples ground-truth **y** and generates the sequence **a** by referring to the source sentence. Then the sequence **a** can be reduced to the normal sentence **s**, which serves as the tailored reference. Therefore, incorporating **y** within the condition can be interpreted as utilizing the upsampled ground-truth.\
> &ensp;&ensp;&ensp; Eq (7) serves as the training objective during the pre-training of the tailor. It enables the tailor to initially learn how to construct ground-truth while considering both the upsampled ground-truth and the source sentence, which is a relatively straightforward task. This establishes a favorable initial state for the SiMT model during the RL fine-tuning stage.\
> &ensp;&ensp;&ensp; We will replace the CTC with dynamic programming and polish the parts involving Equation (5-7) according to your suggestions.
>
> &ensp;
>
> Q6: Lack of details for replication?\
> A6: We apologize for any confusion. Due to space constraints, we mention in Section 4.2 that unlisted experimental details are maintained consistent with [2]. The experimental settings are outlined in the table below.
>
> | Hyperparameter | WMT15 De->En | WMT16 En->Ro | IWSLT15 En->Vi |
> | :-----| :----: | :----: | :----: |
> | model | Transformer Base | Transformer Base | Transformer Small |
> | dropout | 0.3 | 0.3 | 0.1 |
> | optimizer | adam | adam | adam |
> | adam-$\beta$ | (0.9, 0.98) | (0.9, 0.98) | (0.9, 0.98) |
> | lr | 5e-4 | 5e-4 | 5e-4 |
> | weight decay | 1e-4 | 1e-4 | 1e-4 |
> | max tokens | 8192$\times$4 | 8192$\times$4 | 16000 |
>
> &ensp;&ensp;&ensp; We will supplement the experimental settings in the next version and make our code publicly accessible to facilitate better reproducibility..
>
> [2] Ma et al., 2020. Monotonic multihead attention. In ICLR2020.
>
> &ensp;
>
> Q7: L67-L70 appears to be circular reasoning?\
> A7: We are sorry for causing confusion to you. In this context, it means that employing a single reference could cause SiMT models trained at different latency to adopt the identical translation policy.  This may lead to forced predictions in SiMT models trained at low latency or introduce additional delays for SiMT models trained at high latency.  We will follow your suggestions to polish this section in the paper.
>
> &ensp;
>
> Q8: Appendix A highly resembles the content of Section 4 of [Chen+ EMNLP21], which should be omitted?\
> A8: We appreciate your valuable suggestions. [1] make great contributions to SiMT, introducing concepts like HR and AR along with detailed calculation methods. In accordance with your recommendations, we will make modifications and link the relevant content in our paper to [1].
>
> [1] Chen et al., 2021. Improving simultaneous translation by incorporating pseudo-references with fewer reorderings. In EMNLP2021.
>
> &ensp;
>
> Q9: The tables need to be clear. Which datasets are they based on?\
> A9:  We are sorry for causing you to misunderstand. Due to space limitations, we mention in L448-L450 that the experiments conducted in section 5 are all based on the WMT15 De->En dataset. We will include the dataset used for the experiment in the caption of each table according to your reminder.
>
> &ensp;
>
> Q10: Why is 1.89 highlighted in Table 3?\
> A10: The value of 1.89 is highlighted in Table 3 to indicate that, with $\alpha$ set to 0.2, the SiMT model achieves the best trade-offs between latency and translation quality.
>
> &ensp;
>
> Q11: What is Nt in Figure 2?\
> A11: I apologize for any confusion that may have arisen. In Figure 2, Nt represents the number of layers in the tailor. To provide a more comprehensive understanding, We will clearly explain this issue in the main text.
>
> &ensp;
>
> If our answers can solve your questions and eliminate your concerns, we would appreciate it if you could increase the score.

---

### Official Review · Reviewer_WzVC · 2023-08-11

**Soundness:** 4

**Excitement:**

4: Strong: This paper deepens the understanding of some phenomenon or lowers the barriers to an existing research direction.

**Paper Topic And Main Contributions:**

This paper addresses a key difficulty in training simultaneous translation models: the fact that reference translations may have a significantly different word order from the source, forcing the model to guess when translating with low latency. It introduces a new non-autoregressive decoder to generate a custom paraphrase of the target during training to address this. This new module (the "tailor") is learned using reinforcement learning which balances between producing something like the original reference, and something more similar to the "non-anticipatory reference" generated by the translation model using the same latency policy. The experiments are convincing and show that this approach leads to state-of-the-art results (especially for the fixed latency policy, wait-K).

**Questions For The Authors:**

- Any explanation / hypothesis as to why the approaches using the tailor perform so much better on English to Vietnamese than other directions, even beating full-sentence translation? This seems like a major anomaly.


**Reasons To Accept:**

- The authors' approach seems to represent a new state of the art for simultaneous translation.
- The specific approach of using reinforcement to train a "tailor" module to dynamically produce translation targets with semantic ordering more similar to the input is quite novel.
- The analysis seems sound, particularly in demonstrating that the tailor approach does indeed produce better translation targets than the "non-anticipatory references"

**Reasons To Reject:**

- This model seems to introduce a lot of training complexity (and training computational costs) for improvements which are modest, especially in the adaptive policy setting. No discussion or details on the additional training costs are provided.

**Reproducibility:**

4: Could mostly reproduce the results, but there may be some variation because of sample variance or minor variations in their interpretation of the protocol or method.

**Reviewer Confidence:**

2: Willing to defend my evaluation, but it is fairly likely that I missed some details, didn't understand some central points, or can't be sure about the novelty of the work.

---

> ### Author Rebuttal · Authors · 2023-08-27
>
> Thanks for your valuable and insightful comments.
>
> &nbsp;
>
> Q1: Discussion or details on the additional training costs?\
> A1: Our method aims to provide tailored references to the SiMT model online, effectively avoiding forced predictions while ensuring high quality. The experimental results in Figure 5 demonstrate that **our method significantly outperforms previous offline methods**, achieving better latency-quality trade-offs.\
> &ensp;&ensp;&ensp; Regarding the training complexity, our method introduces two additional training processes: pre-training of the tailor and RL fine-tuning. However, it is important to note that **the associated training costs for these processes are relatively modest**. During the pre-training stage, the optimization objective of the model is relatively straightforward, allowing convergence within just five epochs. Furthermore, after achieving a favorable initial state, RL fine-tuning requires only seven epochs of joint training with the SiMT model.\
> &ensp;&ensp;&ensp; Considering the performance enhancements achieved, **the training costs introduced by our method are deemed acceptable**. We acknowledge your concern about training costs and will supplement more details in the next version.
>
> &nbsp;
>
> Q2: Why the approaches using the tailor perform better on English to Vietnamese than other directions, even beating full-sentence translation?\
> A2: **English and Vietnamese belong to distinct language families, making translation between them challenging**. In comparison to other translation directions, translating from English to Vietnamese is inherently more difficult. Our approach employs the tailor to convert ground-truth into tailored references, aligning better with the expression patterns of the source sentence. As a result, our method yields a more significant performance improvement in the En->Vi direction, surpassing translations within the same language family, such as De->En and En-Ro.\
> &ensp;&ensp;&ensp; Prior methods like HMT have showcased the SiMT model can surpass full-sentence translation performance under high latency. This indicates that **a precise policy can guide the translation model to focus on the relevant source tokens while excluding irrelevant ones**. Through the utilization of tailored references, our approach trains the model to acquire a more accurate policy, diminishing forced predictions, and ultimately achieving full-sentence performance.
>
> &nbsp;
>
> If our answers can solve your questions, we would appreciate it if you could increase the score.

---

### Official Review · Reviewer_NbPY · 2023-08-12

**Soundness:** 4

**Excitement:**

4: Strong: This paper deepens the understanding of some phenomenon or lowers the barriers to an existing research direction.

**Paper Topic And Main Contributions:**

This paper introduces a method aimed at addressing the issue of training a low-latency translation model while avoiding ground truth anticipations. It proposes to generate a tailored reference to accommodate the expected latency associated with the simultaneous machine translation (SiMT) problem. It introduces a three-stage method: 1) train a transformer-based SiMT model using ground truth 2) train an additional shallow transformer decoder as the tailor module using ground truth 3) use reinforcement learning method to fine-tune the tailor module. The generated tailored references are used for fine-tuning SiMT. The experiments evaluate the training strategy across three translation tasks and can be worked with different policies (Walk-$k$, and HMT).

Contributions: an effective training mechanism for training a translation model with low-latency, NLP engineering experiment

**Questions For The Authors:**

1. I am not familiar with reinforcement learning, but what is the action and policy of this tailor model? Also, providing a plot of the reward history curve would be helpful.
2. Could you explain more on average lagging (x-axis of Figure 3,4,5), non-anticipatory references, and test-time wait-$k$ methods? It would be useful to include more details about them in the paper.
3. The paper misses some training details (number of epochs, learning rate) in the experimental section. Also, how to input $k$ when training the model?
4. Providing more examples of tailored references would be more helpful.


**Reasons To Accept:**

1. The motivation is clear and the problem is interesting. The paper is well-written mostly.
2. An effective method to automatically generate references for training a low-latency translation model while avoiding forced anticipations.
3. Extensive experimental results and analysis show that their proposed algorithm improves the translation quality, anticipating rate, and hallucination rate for this simultaneous machine translation problem.

**Reasons To Reject:**

1. The RL fine-tuning part of the training method (section 3.2) misses some details and is hard to follow. See Q1.
2. It is unclear how would the unsuccessful tailored references impact the translation performance.


**Reproducibility:**

3: Could reproduce the results with some difficulty. The settings of parameters are underspecified or subjectively determined; the training/evaluation data are not widely available.

**Reviewer Confidence:**

3: Pretty sure, but there's a chance I missed something. Although I have a good feel for this area in general, I did not carefully check the paper's details, e.g., the math, experimental design, or novelty.

---

> ### Author Rebuttal · Authors · 2023-08-27
>
> Thanks for your detailed and valuable feedback.
>
> &ensp;
>
> Q1: What is the action and policy of the tailor model?\
> A1: Reinforcement learning guides the tailor in generating references that meet latency requirements and are of high quality. The action is the output of the tailor, denoted as **a** in the paper. The policy for generating actions employs a greedy policy. Following your suggestions, we will add the plot of the reward history curve in the next version.
>
> &ensp;
>
> Q2: Could you explain more on average lagging, non-anticipatory references, and test-time wait-*k* methods?\
> A2: **Average Lagging (AL)** [1]  is the most widely used latency metric, which measures the degree to which the evaluated policy lags behind the ideal policy (wait-*0* policy). For example, when the AL is equal to 1, it signifies that the evaluated policy lags behind the ideal policy by one token on average.\
> &ensp;&ensp;&ensp; In order to guide the tailored references to exclude hallucinations, **non-anticipatory references**, which are generated by the **test-time wait-*k*** method, serves as the optimization direction for tailor. The **test-time wait-*k*** method implements the wait-*k* policy on a conventional full-sentence translation model. Because complete source content is always available during training, the translations generated by the  **test-time wait-*k*** method are unlikely to hallucinate. As a result, they are referred to as **non-anticipatory references** [2].\
> &ensp;&ensp;&ensp; We will follow your suggestions to include more explanations about AL, tailored references, and test-time wait-*k* method in the next version.
>
> [1] Ma et al., 2019. STACL: Simultaneous Translation with Implicit Anticipation and Controllable Latency using Prefix-to-Prefix Framework. In ACL2019.\
> [2] Chen et al., 2021. Improving simultaneous translation by incorporating pseudo-references with fewer reorderings. In EMNLP2021.
>
> &ensp;
>
> Q3:  Miss some training details in the experiment section?
> A3: I apologize for any misunderstanding I may have caused. Due to space limitations, we mention in Section 4.2 that experimental details not listed are kept consistent with those of [3]. The experimental settings are presented in the following table:
>
> | Hyperparameter | WMT15 De->En | WMT16 En->Ro | IWSLT15 En->Vi |
> | :-----------------------| :---------------------------: | :---------------------------: | :---------------------------: |
> | model | Transformer Base | Transformer Base | Transformer Small |
> | dropout | 0.3 | 0.3 | 0.1 |
> | optimizer | adam | adam | adam |
> | adam-$\beta$ | (0.9, 0.98) | (0.9, 0.98) | (0.9, 0.98) |
> | lr | 5e-4 | 5e-4 | 5e-4 |
> | weight decay | 1e-4 | 1e-4 | 1e-4 |
> | max tokens | 8192$\times$4 | 8192$\times$4 | 16000 |
>
> &ensp;&ensp;&ensp; We will supplement the experimental settings in the next version and make our code publicly available to enhance reproducibility.
>
> [3] Ma et al., 2020. Monotonic multihead attention. In ICLR2020.
>
> &ensp;
>
> Q4: How to input k when training the model?\
> A4: *k* is a hyperparameter of the wait-*k* model, which is manually set based on latency requirements. During training, it enables each target token to attend to different source information by utilizing a cross-attention mask. During inference, it controls the number of source tokens input to the model to achieve simultaneous machine translation.
>
> &ensp;
>
> Q5: Provide more examples of tailored references?\
> A5: By analyzing the tailored references for wait-*1* policy, we observe that our method is largely capable of generating references that meet the requirements. Additionally, we choose some cases and analyze their impact on the SiMT model.
> * Case study of #70 in De->En test set. The tailored reference is generated for wait-*1* policy.
>
> |Sentence|Example|
> | :------------------------| :-------------------------------------------------------------------------------------------------------------------------- |
> | Source Sentence | Die Forscher untersuchten die medizinische Versorgung der Frauen über sieben Jahre . |
> | Ground-Truth | The researchers tracked the women &apos;s medical journeys across seven years . |
> | Tailored Ref | the researchers tracked the women &apos;s medical journeys across seven years . |
> | Translation | the researchers studied the effects of medical care for women over seven years . |
>
> &ensp;&ensp;&ensp; In this example, the tailored reference is identical to ground-truth, yet the SiMT model still maintains high translation quality. This is because the word order of the source sentence closely aligns with that of the ground-truth, resulting in rare forced anticipations under the wait-*1* policy. Consequently, the tailored reference remains the same as the ground-truth.
> * Case study of #401 in De->En test set. The tailored reference is generated for wait-*1* policy.
>
> |Sentence|Example|
> | :-------------------------| :------------------------------------------------------------------------------------------------ |
> | Source Sentence | Ich frage ihn, ob er je wieder eine Stand-up-Comedytour machen wird . |
> | Ground-Truth | I ask whether he will ever do another stand @-@ up tour . |
> | Tailored Ref | I ask him whether he will ever do another stand @-@ up tour . |
> | Translation | I wonder him , whether he will ever be able to get a stand @-@ up tour . |
>
> &ensp;&ensp;&ensp; In comparison to the ground-truth, the tailored reference includes the object 'him', aligning more closely with the conventions of English expression. Consequently, the tailored reference enhances both the fluency and accuracy of translation.\
> &ensp;&ensp;&ensp; According to your suggestions, we will add the examples in the appendix to deepen the understanding of our method.
>
> &ensp;
>
> If our answers can solve your questions, we would appreciate it if you could increase the score.

---

### Meta-Review · Area_Chair_7dfW · 2023-09-16

**Recommendation:** 2

**Metareview:**

The paper addresses simultaneous machine translation (SiMT), which performs translation based on partial input and is useful for real-time translation. A trained SiMT is prone to suffer from the anticipation problem, i.e., the training loss encourages the SiMT to hallucinate. This paper addresses the problem by training from a wait-k non-autoregressive model, which tends to generate less anticipating samples (better aligned with the source order).

Overall, the idea is analogous to using KD to address the multimodality issue of NAT -- I am not saying the paper is the same as KD-NAT, but they share the same spirit.

Reviewers generally find the idea makes sense and intuitively it may mitigate the anticipation problem of SiMT.

However, the paper has a few issues.
1. The paper lacks clarity and some formulations are sloppy. For example, p(y|x,y) is not sound.
2. Some design choices are not clearly explained, such as why choosing CTC-NAT as the tailored reference. In the author response, the author suggest it's for efficiency concerns, but training efficiency is not analyzed in the paper.
3. Experimental details are unclear, especially the RL part, which may hinder the replication of this paper.

---

### Decision · Program_Chairs · 2023-10-07

**Decision:**

Accept-Findings

**Comment:**

The paper addresses simultaneous machine translation (SiMT), which performs translation based on partial input and is useful for real-time translation. A trained SiMT is prone to suffer from the anticipation problem, i.e., the training loss encourages the SiMT to hallucinate. This paper addresses the problem by training from a wait-k non-autoregressive model, which tends to generate less anticipating samples (better aligned with the source order).

Overall, the idea is analogous to using KD to address the multimodality issue of NAT -- I am not saying the paper is the same as KD-NAT, but they share the same spirit.

Reviewers generally find the idea makes sense and intuitively it may mitigate the anticipation problem of SiMT.

However, the paper has a few issues.
1. The paper lacks clarity and some formulations are sloppy. For example, p(y|x,y) is not sound.
2. Some design choices are not clearly explained, such as why choosing CTC-NAT as the tailored reference. In the author response, the author suggest it's for efficiency concerns, but training efficiency is not analyzed in the paper.
3. Experimental details are unclear, especially the RL part, which may hinder the replication of this paper.